# Improving the detection and management of common non-communicable diseases in adults in rural Sidama National Regional State, Ethiopia: Study protocol

**Melaku Haile Likka**◉°*, **Hiwot Abera Areru**◉°, **Betelihem Eshetu Birhanu**°, **Desalegn Tsegaw Hibistu**°, **Bernt Lindtjørn**°

School of Public Health, College of Medicine and Health Sciences, Hawassa University, Hawassa, Ethiopia

◉ These authors contributed equally to this work.
* melakuhaile@hu.edu.et

## Abstract

### Background

The World Health Organization (WHO) has designed a package of essential non-communicable disease (PEN) strategies to improve the detection and management of non-communicable diseases. However, the implementation of the PEN in rural health facilities in Sidama National Regional State is at an early stage, and the readiness of rural primary healthcare units (PHCUs) to implement the strategy is unknown. We, therefore, propose to apply the strategy in the catchment areas of Dobe-Toga Health Center, a rural PHCU in Sidama National Regional State, Ethiopia, and improve non-communicable diseases care among adults aged ≥45 years.

### Aim

We primarily aim to evaluate the effectiveness of a WHO PEN-based care model, which we will implement in a rural setting, in controlling blood pressure and glucose levels among older adults. It will also determine the prevalence of undiagnosed hypertension, pre-diabetes, and diabetes. Additionally, the study seeks to assess the readiness of rural PHCUs to implement the WHO PEN approach and examine the relationship between NCD diagnoses and community-based health insurance (CBHI) membership.

### Methods

The study will be conducted in the catchment areas of Dobe-Toga Health Center from April 2024 to February 2025 and includes multiple components. One component is a cross-sectional readiness assessment of PHCUs in Sidama National Regional State, which will be undertaken involving 41 health centers and 4 primary hospitals,

**Data availability statement:** No datasets were generated or analyzed during the current study. All relevant data from this study will be made available upon study completion.

**Funding:** The studies will be funded by the South Ethiopia Network of Universities in Public Health II (SENUPH II), which receives funding from the Norwegian Programs for Capacity Development in Higher Education and Research for Development (NORHED); SENUPH II NORHED [1326000]. The funding will be awarded to B.L.

**Competing interests:** The authors have declared that no competing interests exist.

**Abbreviations:** ADA, American Diabetes Association; BP, Blood pressure; CVM, Contingent valuation method; DBP, Diastolic blood pressure; HbA1c, Haemoglobin A1c; LMICs, Low-and-middle-income countries; mmHg, Millimetres of mercury; mmol, Millimole; mol, Mole; NCDs, Non-communicable diseases; PHCUs, Primary healthcare units; SBP, Systolic blood pressure; SNRS, Sidama National Regional State; T2DM, Type-2 diabetes mellitus; WHO, World Health Organization; WHO PEN, World Health Organisation Package of Essential Noncommunicable Disease; WTP, Willingness to pay.

triangulated by qualitative data collection. The remaining study components will be conducted exclusively in the catchment areas of Dobe-Toga Health Center in two phases. In Phase 1, cross-sectional surveys will be conducted to determine the prevalence of undiagnosed hypertension, type 2 diabetes mellitus, and pre-diabetes in a randomly selected sample of 3,301 older adults. Additionally, the participants' willingness to pay (WTP) for HbA1c tests will be assessed, and CBHI-related surveys will be conducted. In Phase 2, the individuals diagnosed with these conditions will be linked to the health center and enrolled in a WHO PEN–based care model. The effects of the care model in controlling blood pressure and glucose will be examined. Furthermore, the participants' adherence to self-care practices will be determined.

## Introduction

### Background

Non-communicable diseases (NCDs), including cardiovascular diseases, cancers, chronic respiratory diseases, and diabetes, are chronic conditions with prolonged duration and complex causes. They are often influenced by genetic, physiological, environmental, and behavioral factors. Globally, NCDs represent a significant burden on public health, contributing substantially to premature mortality and disability [1,2].

Hypertension and diabetes mellitus are two common NCDs that pose a significant health challenge, especially in low- and middle-income countries (LMICs) [3]. According to the World Health Organization (WHO), approximately 1.28 billion adults aged 30–79 have hypertension, with two-thirds residing in LMICs. This figure has risen dramatically from 594 million in 1975 to 1.13 billion in 2015.

The African Region bears a disproportionate burden, with a hypertension prevalence of 27% [4]. Diabetes mellitus (DM) is another major NCD affecting over half a billion adults worldwide, with type 2 diabetes mellitus (T2DM) being the predominant form. The number of adults diagnosed with diabetes is projected to rise to 643 million by 2030 and reach 783 million by 2045 [5]. In Sub-Saharan Africa (SSA), the prevalence of overall diabetes and prediabetes is concerning, with rates of 6.8% and 25%, respectively. The adult population aged 50–59 years is particularly vulnerable, with a reported DM prevalence of 14.9% [6]. Diabetes has emerged as a leading cause of death globally. In 2016, the WHO reported that diabetes was responsible for 1.6 million deaths, half of which occurred in individuals under the age of 70 [7,8].

Ethiopia grapples with a triple health challenge, encompassing NCDs, injuries, and infectious diseases [9]. The 2018 Ethiopian NCD and Injury Commission Report revealed that NCDs share 37.5% of the disease burden [10]. Nearly 40% of deaths in Ethiopia result from NCDs [11], with hypertension and diabetes accounting for over a third of the disease burden and 43% of fatalities [12,13]. Hypertension is a major public health issue in Ethiopia, responsible for more than half of cardiovascular diseases and affecting nearly a quarter of the population [13]. Controlling hypertension reduces the risk of developing other NCDs like heart failure, chronic kidney disease, and stroke [14]. However, only 20% of adults in Ethiopia manage to control their

hypertension [4]. Additionally, the pooled prevalence of uncontrolled hypertension is 48% [15], underscoring a significant gap in diagnosis and treatment.

Similarly, the prevalence of DM was found to be 6.5% in Ethiopia [16], with undiagnosed diabetes ranging from 5.80% to 10.2% [17,18]. The existing studies used less reliable techniques such as fasting and random blood sugar tests to determine the prevalence of undiagnosed diabetes. None of the studies on the prevalence of DM used hemoglobin A1c (HbA1c) tests, which provide an average of blood glucose levels over the past 2–3 months. HbA1c is considered the gold standard for assessing the compensation and treatment of diabetes and for diagnosing DM [19]. Additionally, the American Diabetes Association and the WHO recommend using HbA1c with a cut-off value of 6.5% for the early diagnosis of type 2 diabetes mellitus (T2DM) and pre-diabetes in asymptomatic individuals [20].

The WHO has developed evidence-based recommendations called the Package of Essential Non-communicable Diseases (PEN) to enhance the detection, treatment, and management of NCDs, including diabetes and hypertension. These recommendations consist of affordable medications for treatment, non-pharmacological and pharmaceutical methods for modifying risk factors, and cost-effective strategies for early detection of NCDs [2]. Early detection and management are crucial interventions for preventing complications and deaths related to hypertension and diabetes, as well as for enhancing treatment outcomes. However, nearly half of people with diabetes remain undiagnosed, with type 2 diabetes being the most common form worldwide [5].

Ethiopia has been implementing various measures to address the growing burden of NCDs, such as hypertension and diabetes. These measures include developing a national guideline for early detection and treatment and enhancing the capacity of primary healthcare units (PHCUs) in low-resource settings [2,21–23]. The PHCUs are crucial for addressing NCDs and providing integrated, equitable care. Enhancing the diagnostic capacity for NCDs is one of the strategic initiatives in Ethiopia's current national strategy for preventing and controlling cardiovascular and chronic diseases [24]. However, their effectiveness in Ethiopia is limited by insufficient technical capacity, lack of diagnostic tests, limited support from national authorities, inadequate equipment and medicines, and poor integration of NCD care to the PHCUs [23,25,26]. Additionally, the implementation of the PEN is inadequate in Ethiopia and is still in its early stages in the Sidama National Regional State (SNRS). Besides, the readiness of PHCUs to implement this intervention strategy remains uncertain in this region.

We, therefore, propose to improve the early detection and management of hypertension, pre-T2DM, and T2DM among older adults (aged 45 years and older) in the catchment areas of Dobe-Toga Health Center, a rural PHCU in SNRS, Ethiopia. Hypertension and diabetes were chosen as the focus of this study because of their high prevalence, significant public health impact, and the feasibility of measuring them within our available resources. Both conditions are also prioritized in national and global NCD strategies, which enhances the potential policy relevance of our findings. This will be accomplished through community-based screening and the implementation of a detailed care model as described in Methods section for individuals diagnosed with the above conditions. As part of implementing the care model, we will determine the prevalence of undiagnosed hypertension, T2DM, and pre-T2DM among the study participants. Additionally, we will assess the effectiveness of the care model in reducing blood pressure and hemoglobin A1c levels after six months of implementation. This study will also evaluate the readiness of the primary healthcare system in SNRS to implement the WHO PEN approaches.

Furthermore, we will investigate whether having a family member diagnosed with any of the mentioned conditions will affect the respondents' decision to join the community-based health insurance (CBHI) scheme introduced in Ethiopia in 2014 [27]. We will also assess the participants' willingness to increase the annual premium for membership if the schemes encounter sustainability challenges due to moral hazards and adverse selections. Unlike most schemes in LMICs [28], the Ethiopian CBHI does not require co-payment for service delivery to eligible individuals. Copayment is important in mitigating the impact of moral hazards. We will also examine the willingness of CBHI members to co-pay for healthcare services.

## Study aims

The study primarily aims to evaluate the effectiveness of a WHO PEN-based care model, which we will implement in a rural setting, in controlling blood pressure and glucose levels among older adults in Dobe-Toga PHCU's catchment areas. It will also determine the prevalence of undiagnosed hypertension, pre-T2DM, and T2DM in the target population. Furthermore, the study will assess the readiness of rural PHCUs to implement the WHO PEN disease intervention approach in SNRS and examine the relationship between diagnoses with NCDs and CBHI membership.

## Objectives of the study

The study is structured into two distinct categories, each with its specific objective(s):

a) Studies on undiagnosed hypertension and T2DM

1. To determine the prevalence of undiagnosed hypertension, pre-T2DM, and T2DM (confirmed using HbA1c) among the adults in the catchment areas of Dobe-Toga Health Center, Shebedino District, SNRS.

2. To evaluate the effect of the WHO PEN-based hypertension and T2DM management and care package on blood pressure and blood glucose levels among the study participants.

3. To determine the overall level of self-care practice among hypertensive clients enrolled in the WHO PEN care model and identify factors associated with these practices.

4. To assess the willingness to pay for HbA1c testing among the adult population in the study area.

b) Other studies

5. To assess the readiness of PHCUs in SNRS to implement the WHO PEN intervention.

6. To compare CBHI enrollment among families with and without members diagnosed with NCDs.

7. To determine the willingness for co-payment for health services among the CBHI members in the study area.

8. To assess the willingness to raise the annual CBHI membership premiums as an option to sustain the scheme in the study area.

## Methods and designs

### Study setting and period

The study will be conducted from April 2024 to February 2025 in the catchment areas of Dobe-Toga Health Center. This rural PHCU is located in Shebedino District, one of the 36 districts in the SNRS, with Leku Town as its capital. Leku Town is about 300 kilometers south of Addis Ababa, the nation's capital, and 24 km from Hawassa, the capital city of SNRS. According to the Ethiopian Central Statistical Agency's population projections for 2021/2022, the district's total population was 204,618, with 100,263 males and 104,355 females.

Shebedino District comprises 23 rural and nine urban kebeles (the smallest administrative unit in Ethiopia). The district has six health centers, including Dobe-Toga Health Center. Dobe-Toga Health Center, together with its four satellite health posts, serves a population of 38,874 people across its four catchment kebeles—Gonowa-Gabalo, Dobe-Toga, Howolso, and Gobe-Hebisha. Based on the age distribution in SNRS, individuals aged 45 and above make up 13% of the population, totaling 5,054 people in these four kebeles.

### Study designs

The study will proceed in two phases. In Phase 1, community-based cross-sectional surveys will be conducted to identify individuals with undiagnosed hypertension, pre-T2DM, and T2DM. In Phase 2 of the study, these individuals will be

enrolled in a follow-up program at Dobe-Toga Health Center, where they will receive the WHO PEN-based care model described below. The care model will be implemented for 6–8 months and completed by February 2025.

Additionally, surveys related to CBHI will be conducted on the same study population during the initial survey, and enrollment status will be reassessed at the end of the follow-up to determine if the diagnosis affects households' decisions to enroll in the schemes. Follow-up tests and examinations for the other studies will be conducted in the 1st and 3rd months of enrollment in the care model, and the final follow-up data will be collected at the end of the 6th month of enrollment. The second phase of the research will utilize a pre-post design.

Furthermore, a cross-sectional survey of samples of PHCUs (health centers and primary hospitals), triangulated with qualitative data, will be employed to assess the readiness of the PHCUs in SNRS to implement the WHO PEN interventions (Objective 5). The quantitative data for Objective 5 will be collected using an observation checklist focused on NCD management and care inputs from randomly selected health facilities within the SNRS. The qualitative data for triangulating the cross-sectional survey will be obtained through in-depth interviews of NCD focal persons of selected PHCUs and healthcare authorities in the SNRS Health Bureau and selected lower-level administrative organs.

## Populations

For the community-based cross-sectional studies, the source population will include all adults aged 45 or older residing in the catchment areas of Dobe-Toga Health Center. The follow-up focuses on adults from this same population who will be diagnosed with elevated blood pressure or diabetes mellitus during the cross-sectional survey. For the cross-sectional surveys of PHCUs, all health centers and primary hospitals in the SNRS will be considered the source population.

The population for the qualitative data supplementing the quantitative data for Objective 5 will consist of healthcare authorities at the regional, zonal, and district levels, as well as NCD focal persons at selected health administration organs and health facilities.

## Eligibility criteria

For the community-based cross-sectional studies, eligible participants will be adults aged 45 or older who have resided in the selected kebeles for at least six months. The cohort studies include individuals diagnosed with hypertension, pre-T2DM, and T2DM based on initial screening with an FBS test and subsequent confirmation with an HbA1c test. All health centers and primary hospitals in the SNRS will be eligible for the PHCU surveys. Individuals with severe cognitive challenges, disabilities, or pregnancy will be excluded from the study due to the potential temporary nature of hypertension or diabetes during these physiological states.

## The hypertension and diabetes care model package

The intervention for the pre-post study at Dobe-Toga Health Center is the 'Hypertension and Diabetes Care Model.' This model includes community-based screening for hypertension, pre-T2DM, and T2DM, staff training to enhance the health center's capacity, accurate diagnosis of these conditions, improved patient engagement, and a robust follow-up system. The follow-up system will involve clinical and laboratory-based management, medication, and behavioral counseling at the point of care. The care model aims to improve the detection and management of hypertension and diabetes, enhance quality of life, increase adherence to the care model, foster patient ownership, and improve diabetes control.

**Community-based screening for hypertension, pre-diabetes, and diabetes.** To assess the prevalence of undiagnosed hypertension, pre-T2DM, and T2DM within the elderly population, we will conduct the following activities: (1) identifying the target population and specific locations for the screening program; (2) developing a screening protocol that includes measuring blood glucose levels and collecting relevant clinical and demographic data; (3) recruiting and training the PHCU's health professionals and data collectors to conduct the screenings and providing them with necessary equipment and supplies; (4) carrying out the screening program within the community, reaching as many eligible individuals as possible; and (5) collecting and analyzing the screening data to identify the prevalence of undiagnosed

hypertension, pre-T2DM, and T2DM, identify associated risk factors, and linking individuals with T2DM and hypertension to Dobe-Toga Health Center for further management and care.

**Strengthening the capacity of the health center to detect hypertension, pre-T2DM, and T2DM by training the staff.** To enhance the capabilities of the targeted PHCU in detecting hypertension and T2DM, the following activities will be implemented:(1) Developing a training document that covers the fundamentals of hypertension and diabetes detection and management. This document will align with current guidelines, including the WHO-PEN intervention approach [2], best practices, and evidence-based recommendations; (2) Identifying and selecting trainers with expertise in diabetes and hypertension care to serve as trainers, and recruiting trainees; (3) Conducting the training sessions using a blend of classroom instruction, hands-on training, and mentoring; (4) Providing ongoing support, including refresher training for PHCU professionals to ensure that they effectively detect and manage hypertension and diabetes; (5) Monitoring and evaluating the progress of the PHCU professionals in detecting and managing hypertension, and T2DM. By undertaking these activities, we aim to significantly improve the capacity of health center staff in managing these conditions.

**Proper diagnosis of hypertension, pre-T2DM, and T2DM.** To effectively diagnose hypertension, pre-T2DM, and T2DM as part of the care model, the following procedures will be implemented: (1) Obtain a detailed patient history: Collect information about symptoms and risk factors associated with hypertension, pre-T2DM, and T2DM. This includes inquiries about family history, sedentary lifestyle, overweight/obesity, age, and other relevant factors; (2) Perform blood pressure and blood glucose assessments: Measure blood pressure and conduct blood glucose tests to identify whether the patient is hypertensive, hyperglycemic, hypoglycemic, pre-hyperglycemic, or within normal ranges. An FBS test will be used to screen for possible cases of pre-T2DM and T2DM; (3) Confirm possible cases with HbA1c testing: Use the HbA1c test to confirm any suspected cases of pre-T2DM and T2DM; (4) Conduct additional biochemical tests: Perform further tests, including urine tests, lipid profiles, kidney function tests, and foot examinations, to assist in treatment and monitor for complications. These steps will ensure a comprehensive approach to diagnosing and managing these conditions.

**Improving patient ownership in hypertension and diabetes management.** Enhancing patient ownership, also known as patient engagement or self-management, is crucial for effectively managing hypertension, pre-T2DM, and T2DM. By focusing on a person-centered approach, we will incorporate the following strategies into the care package to improve patient ownership:

i. Providing accurate information: The participants in the care model will receive comprehensive information about hypertension, pre-T2DM, and T2DM, along with their management strategies. This helps them understand their conditions and actively participate in their care.

ii. Setting goals: The participants will be encouraged to establish specific, measurable, and achievable goals related to their condition management, helping them stay motivated and focused.

iii. Personalized care: The care is tailored to the individual needs, preferences, and values of patients, fostering trust and enhancing communication between patients and healthcare providers.

iv. Shared decision-making: The participants in the Care Model will be involved in decisions about their care, making them feel more invested in the process and increasing the likelihood of adherence to treatment plans.

v. Regular monitoring: The participants will be encouraged to regularly monitor their blood glucose and pressure levels, as well as other condition-related parameters, allowing them to identify patterns and adjust their treatment plans as necessary.

vi. Building Community Support: The participants will be connected with others who have T2DM, pre-T2DM, or hypertension, providing a sense of community and support that is beneficial for improving self-management.

These strategies collectively aim to empower patients to manage their health more effectively through enhanced self-management, communication, and education [29].

## Sample size determination

The sample size for the community-based cross-sectional surveys was calculated using OpenEpi version 3.01 [30]. The following parameters were used: p = 33.3% (based on the weighted prevalence of undiagnosed hypertension in Wolaita Zone, Ethiopia, which was 29.8% [26.5%−33.3%]) [31], a margin of error of 2%, a design effect (DEFF) of 2, a confidence level (CL) of 95%, a non-response rate (NRR) of 10%, and an adjusted finite population correction (N = 5054). The calculated sample size was 3301. All the community-based survey (prevalence of hypertension, pre-T2DM, T2DM, comorbidity of both T2DM and hypertension and CBHI) data will be collected from these individuals.

In the second phase of the pre-post-study design at the health center, all individuals diagnosed with hypertension, pre-T2DM, and T2DM will be followed up and included in the studies.

For the quantitative data on PHCU readiness to implement WHO PEN interventions in the SNRS, a 30% sample of PHCUs will be selected. Sidama National Regional State has a total of 150 PHCUs—137 health centers and 13 primary hospitals. Among them, 41 health centers and four primary hospitals (45 PHCUs) will be selected randomly and included in the study.

The qualitative data to triangulate the readiness findings will be obtained via in-depth interviews with purposively selected informants capable of providing rich and detailed information on the readiness of primary healthcare units to implement the WHO PEN strategy in the Sidama Region. These informants will include regional, zonal, and district health-care administrative experts, as well as NCD focal persons at selected administrative organs and health facilities. Interviews will be conducted until information saturation is reached, defined as the point at which no new themes or insights emerge. We estimate that this saturation will be achieved within 10–15 in-depth interviews.

## Sampling procedure and follow-up schedule

A multi-stage sampling technique will be applied to select the study participants. First, Dobe-Toga Health Center, a rural PHCU with four catchment kebeles, was chosen based on its total population size and rural location among the district's six health centers. The health center's catchment kebeles are Gonowa-Gabalo, Dobe-Toga, Howolso, and Gobe-Hebisha, with a combined population of 38,874 in 7,933 households.

The total sample will be allocated to these kebeles proportionally to their population sizes. Households with eligible study participants will be selected using systematic sampling based on a household list. The systematic sampling interval (K) will be determined by dividing the total target population by the estimated sample size, and then every Kth household will be selected (Fig 1). When a selected household has multiple eligible respondents, all will be included.

Following the selection of the kebeles, a cross-sectional study will be conducted to determine the burden of hypertension, pre-T2DM, and T2DM among the participants. Those diagnosed with hypertension will be confirmed positive without further assessment, while those identified with pre-T2DM and T2DM will be confirmed using an HbA1c test. Individuals diagnosed with hypertension and T2DM (ascertained via HbA1c) will be referred to Dobe-Toga Health Center for ongoing care and the application of the care model for 6–8 months.

Additionally, participants will be surveyed about their willingness to pay (WTP) for HbA1c tests for diabetes diagnosis, their participation in community-based health insurance (CBHI), and their WTP for copayments and increased annual premiums to sustain the CBHI program in their community. To compare CBHI-related decisions after the diagnosis with hypertension, pre-T2DM, and/or T2DM, participants will be asked about CBHI items again at the end of the follow-up.

## Study variables

**The outcome variables.** NCDs care and management input and process indicators of health centers and primary hospitals, systolic and diastolic blood pressure and blood glucose level during the community-based screening, 3rd and 6th months of intervention and follow-up; HbA1c during community-based screening (for those whose FBS ≥ 5.6 millimole/liter

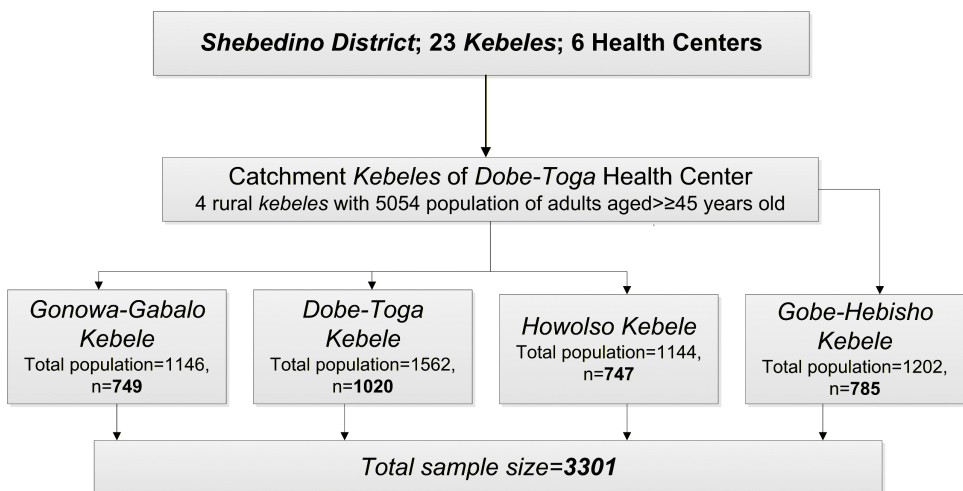

**Fig 1. Schematic presentation of the sampling procedure.**

(mmol/L) of blood only), 3rd and 6th month follow-up; self-care practice; adherence to the WHO PEN-based care model; WTP for HbA1c for diagnosis of T2DM during the community-based screening; enrolment status of the community-based health insurance (during the community-based screening and at the end of the follow-up); willingness to renew CBHI membership; willingness to raise the annual premium for CBHI membership; willingness to pay the co-payment during healthcare services.

**The exposure variables.** Locations of the facilities to be assessed for the readiness of the PHCUs; type of the facilities; managing authorities of the health facilities; area of the facility (urban/rural); types of services provided; socio-demographic and economic characteristics (sex, age, marital status, educational level, occupation, income/wealth status); self-rated health status; family size; wealth index of the households; distance to the health facility, behavioral risk factors (tobacco and khat use, alcohol consumption); dietary behaviors (fruit and vegetable intake, salt intake, and physical activity); history of NCDs; physical measurements (height, weight, body mass index, cholesterol level, and blood pressure).

## Operational definitions

i)   The readiness of the PHCUs to implement the WHO PEN disease interventions will be assessed using the WHO's Service Availability and Readiness Assessment (SARA) methodology [32] and the WHO PEN Interventions for Primary Healthcare reference manuals [2].

ii)  Hypertension is defined as an average blood pressure (BP) measurement, taken over three readings, of >140/90 mmHg, or >130/80 mmHg for patients with diabetes or chronic kidney disease [34]. Prior to measuring the BP, the data collectors will ensure the following procedures are adhered to:

- The patient rests for at least 5 minutes prior to the measurement and avoids alcohol, caffeine, and physical activity;

- A properly calibrated automatic BP monitor (Omron HEM 7080 BP apparatus) will be used;

- The patient will sit with their back supported and arm positioned at heart level;

- Three consecutive BP readings will be taken at least five minutes apart with an appropriate-sized cuff

- Blood pressure readings will be recorded and averaged to statistically determine the individual's blood pressure status

iii) The self-care practices of hypertensive clients on follow-up will be assessed using the Hypertension Self-Care Activity Level Effects (H-SCALE) at the end of the follow-up. This 31-item scale evaluates self-care aspects, including medication adherence, dietary management, smoking status, physical activity, weight management, and alcohol intake [33,34]. Self-care practices will be categorized as good (adherence to all the components of the H-SCALE) and poor (non-adherence to at least one element of the H-SCALE).

iv) Type 2 diabetes mellitus will be confirmed based on the following criteria: a positive history of diabetes mellitus, the use of anti-diabetic medication, or fasting blood sugar (FBS) levels measured during community-based screening. The HbA1c test will be used to confirm the diagnosis at the rural PHCU. The HbA1c test evaluates the average blood glucose levels in the red blood cells over the previous 2–3 months and it is used to diagnose and monitor diabetes, as well as assess the risk of developing diabetes and other long-term complications [35]. According to the American Diabetes Association (ADA), pre-diabetes (Pre-DM) will be considered if an FBS between 5.6 mmol/L and 6.9 mmol and HbA1c 39–47 mmol/mol while T2DM will be confirmed if the FBS ≥ 7.0 mmol/L and the HbA1c ≥ 48 mmol/mol. The HbAlc test is preferred over other diagnosis methods due to its superior advantages, such as convenience (no fasting is required), greater pre-analytical stability, and fewer day-to-day fluctuations during periods of stress, dietary changes, or illness [36,37].

v) To assess CBHI membership among families with and without members affected by NCDs, a survey related to CBHI will be administered to study participants at two points: during the initial community-based survey and at the end of the follow-up period. During the initial assessment, we will examine the level of CBHI enrollment and renewal among the entire study population. In the second phase, individuals diagnosed with hypertension, T2DM, or/and pre-T2DM will be asked whether they plan to enroll in CBHI or renew their membership during the next recruitment phase for the scheme. Participants who respond "no" about their intentions to enroll or renew will be classified as unwilling to join or continue their membership. Conversely, those who respond "yes" will be classified as willing to become CBHI members or to renew their membership. Finally, we will compare the willingness to join or renew membership between those with family members affected by NCDs and those without.

vi) To assess the WTP for additional premiums for annual CBHI membership and copayment for medical services, we will employ the double-bound dichotomous choice-contingent method (DBDC-CVM), one of the direct methods used to elicit individuals' WTP for public and non-market goods or services [38]. This direct elicitation technique is effective for determining individuals' WTP for public and non-market goods or services, as it is information-intensive, asymptotically more efficient, and less susceptible to starting point and strategic biases, thereby reducing the necessity for a large sample size [38,39]. Suppose that an individual's indirect utility is contingent upon purchasing a health insurance policy and income y, let q1 and q0 measure the level of utility associated with having and not having health insurance, respectively. The WTP is the amount of money an individual is willing to pay as a premium. X denotes the vector of other factors, such as age, sex, education, and health status, that may affect the individuals' preferences. π represents the perceived probability of falling ill, and Ɛ captures other unobservable factors to the researcher. Then, the individuals will opt to purchase the health insurance policy only if v [(q1, y-WTP, X, π) + Ɛ1) ≥ v [(q0, y, X, π) + Ɛ0); where: Ɛ1 and Ɛ0 are random errors distributed independently with mean zero

In DBDC-CVM, each respondent is initially asked if they are willing to pay the first bid amount. If the respondent answers "yes", they are then presented with a second, higher bid, and asked their WTP for this new amount. Conversely,

if the respondent answers "no" to the first bid, a second, lower bid is given, and their willingness to pay is reassessed. Should the respondent answer "no" or "yes" to both the first and second bids, they are asked to specify the maximum amount they are willing to pay, as demonstrated in Fig 2.

For this study, four initial bid amounts for additional annual premiums and co-payments were set for the respondents to choose from using the lottery method. The second higher and lower bids were twice and half of the initial bids chosen, respectively. The initial bid amounts for annual premiums are set at 200, 250, 300, and 400 Ethiopian birrs, while the copayment percentages are set at 10%, 15%, 20%, and 25%. Before eliciting the WTP, hypothetical scenarios will be presented to the respondents. Given that the primary aim of this WTP study is to ensure the sustainability of the CBHI schemes, respondents will be asked to choose between discontinuation of the schemes and an increase in annual premiums and copayments for the services.

vii) A DBDC-CVM will be used to assess the WTP for HbA1c diagnosis of T2DM (Fig 2). Four initial bids (250, 300, 400, and 500 Ethiopian birrs) will be randomly assigned to participants. The second highest and lowest bids will be 150% and 50% of the initial bids, respectively.

## Statistical analyses

For the quantitative data, data collected using KoboToolbox will be exported to R version 4.4.1 for comprehensive statistical analysis. Prior to analysis, a data cleaning process will be conducted to eliminate errors, inconsistencies, and missing values. Subsequently, the main characteristics of the data will be summarized. For the community-based study,

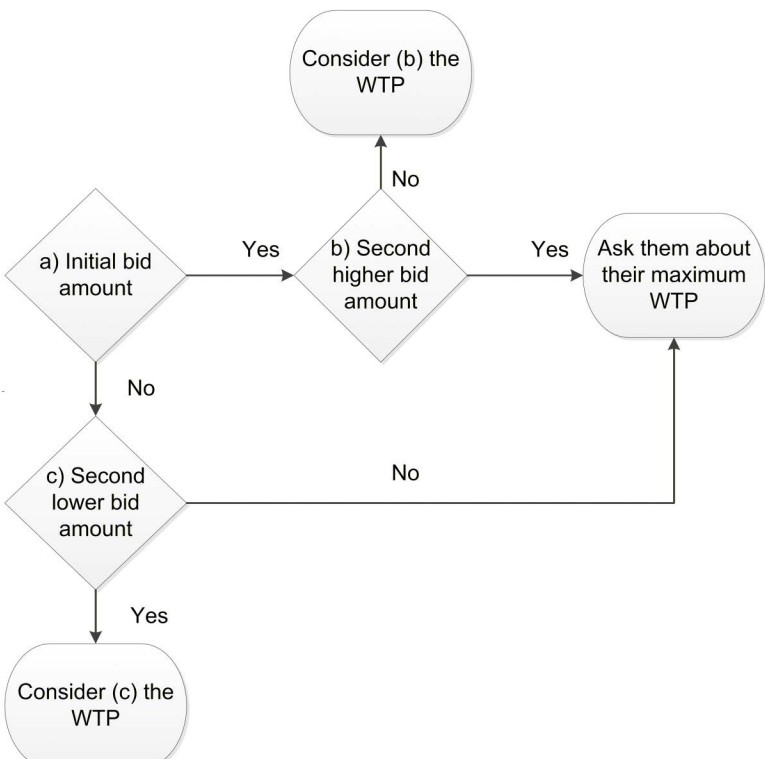

**Fig 2. The double-bound dichotomous choice-contingent valuation method that will be used to elicit WTP for extra premiums and copayment in the study area.**

binary logistic regression will be employed to identify factors associated with the prevalence of undiagnosed hypertension, T2DM, and pre-T2DM.

The change in mean blood pressure, HbA1c, and blood glucose level at 3 and 6 months will be determined using a dependent t-test to measure the effect of the exposure variables on the outcomes. Linear regression will be conducted to identify predictors for improvements in blood pressure, blood glucose levels, and HbA1c changes. The exposure variables will be examined for multicollinearity using variance inflation (VIF). The strength of the association will be reported with 95% confidence intervals (CIs) and the corresponding adjusted odds ratios (AORs). A *p-value* of <.05 will be considered statistically significant. To determine adherence to the anti-hypertensive care, ordinal logistic regression will be utilized.

To determine the relationship between CBHI membership and the covariates as well as the willingness to renew membership, binary logistic regression will be employed. The WTP for additional premiums and copayments, assessed using the DBDC-CVM, will be estimated through seemingly unrelated bivariate probit regression. The R package called "DCchoice" will be used to compute the mean/median WTP and to establish Krinsky and Robb's confidence interval for the mean/median WTP. This method will similarly be applied to determine the WTP for HbA1c tests for diagnosing blood glucose levels. The statistical analysis plans for each study objective are summarized in Table 1.

Qualitative data analysis will be conducted using a thematic analysis approach. All interviews will be audio-recorded, transcribed verbatim, and imported into ATLAS.ti software for data management and analysis. An initial coding framework will be developed based on the study objectives and relevant literature, and transcripts will be systematically coded to identify meaningful segments of text. Codes will be reviewed and refined through an iterative process to ensure consistency and comprehensiveness. Related codes will then be grouped into categories and themes. The resulting themes will be used to explore and interpret the facilitators and barriers to implementing the WHO PEN strategy in rural primary healthcare settings in the SNRS.

**Table 1. Statistical analysis plans for each objective in this research.**

| Objectives | Outcome variable | Statistical analysis |
|---|---|---|
| To determine the prevalence of undiagnosed hypertension, pre-T2DM, and T2DM (confirmed using HbA1c) among the adults in the catchment areas of Dobe-Toga Health Center, Shebedino District, SNRS. | Undiagnosed T2DM, pre-T2DM, and hypertension (diagnosed/not diagnosed) | Binary logistic regression |
| To evaluate the effect of the WHO PEN-based hypertension and T2DM management and care package on blood pressure and blood glucose levels among the study participants. | Blood glucose level, HbA1c levels and systolic and diastolic blood pressure levels (continuous) | Dependent t-test |
| To determine the overall level of self-care practice among hypertensive clients enrolled in the WHO PEN care model and identify factors associated with these practices | Self-care practice (poor/good) | Binary logistic regression |
| To assess the WTP for HbA1c testing among the adult population in the study area | WTP for HbA1c test (bivariate outcome variable) | Seemingly unrelated probit regression |
| To assess the readiness of PHCUs in SNRS to implement the WHO PEN intervention. | Readiness of the PHCUs | Linear regression |
| To compare CBHI enrollment among families with and without members diagnosed with NCDs. | Willingness to enroll in CBHI (yes/no) | Binary logistic regression |
| To determine the willingness for co-payment for health services among the CBHI members in the study area. | Willingness to make copayment (bivariate outcome variable) | Seemingly unrelated probit regression |
| To assess the willingness to raise the annual CBHI membership premiums as an option to sustain the scheme in the study area. | Willingness to raise the annual membership premium (bivariate outcome variable) | Seemingly unrelated probit regression |

## Data collection instruments

The questionnaires used for collecting quantitative data will be adapted from the WHO STEP-wise approach to the NCD surveillance tool, the Service Availability and Readiness Assessment (SARA) methodology, the WHO PEN Interventions for Primary Healthcare reference manuals, and other relevant instruments. These questionnaires will initially be prepared in English and subsequently translated into the local language, Sidaamu Affoo. To ensure consistency and accuracy, bilingual experts will perform a back-translation of the Sidaamu Affoo version into English, followed by a re-translation back into Sidaamu Affoo. The final version will be administered to respondents through face-to-face interviews.

The interview guide for the in-depth interviews (qualitative data to supplement the quantitative data for Objective 5) will be originally drafted and administered in Amharic. However, the data collectors will be bilingual and facilitate interviews in Amharic, Sidaamu Affoo, or a combination of both, based on the informants' preferences.

## Quality assurance

Comprehensive measures will be implemented to ensure the quality of research at every stage, from tool design to data analysis and interpretation. The questionnaires will be administered to respondents through face-to-face interviews using the interviewer-administered Sidaamu Affoo version of the questionnaire. These interviews will be facilitated by the Android-based KoboToolbox platform on smartphones [40].

The tool will be pretested in Remeda Kebele of Habela District, SNRS, with 160 individuals (5% of the sample size). Appropriate modifications will be made based on the pre-test findings. Data collectors and supervisors will be selected based on their education, profession, data collection experience, and language skills. Before starting data collection, they will be trained on the use of the KoboToolbox data collection platform, including experience exchange and practical demonstrations. The data collection process will be closely monitored. The Geographical Positioning System (GPS) on KoboToolbox will be activated to track the location of each household. All questions will be checked for completeness as soon as the data collectors submit the data to the server set up for this study, and immediate actions will be taken if any issues are identified. Proper feedback will be provided to data collectors, and in some kebeles, data collection may be repeated if necessary, involving 5% of the sample size. Advanced techniques and appropriate modeling will be applied during the analysis. Key assumptions and the fitness of the statistical models will be checked using standard procedures.

## Ethical consideration

Ethical clearance for this study was initially obtained from the Institutional Review Board (IRB) of the College of Medicine and Health Sciences, Hawassa University (Ref. No: IRB/342/15) in June 2023. Following modifications to the original protocol, a revised ethical clearance was requested and subsequently approved by the Board (Ref. No: IRB/011/16) in April 2024. Before data collection commences, an official letter of support will be obtained from the SNRS Health Bureau, and formal permission will be sought from the Shebedino District Health Office.

During community-based data collection, participants will be informed about the study's purpose. Written informed consent will be obtained from all participants, with the exception of those involved in the health facility observation (Objective 5). Participants will electronically sign the consent form in the KoboToolbox platform. The system will require a signature before allowing the user to proceed to the survey items. For participants who are unable to read and write, the consent form will be read aloud in their local language, and an 'x' mark will be recorded in KoboToolbox to document their agreement. All participants will be informed about the study's purpose and their right to self-determination will be respected.

To maintain confidentiality and privacy, participants' names will be anonymized, and data collection tools will be password-protected. Participation in the study will be voluntary, and participants who will be unwilling to continue or wish to withdraw at any stage will be free to do so without any restrictions. During data collection, counseling will be provided for newly diagnosed clients. The same ethical guidelines will be followed during the follow-up data collection.

Quantitative data for Objective 5 will be collected using an observation checklist. Given the minimal risks associated with the study, only verbal consent will be required from the authorities of the selected health facilities. These authorities will be informed about the study's purpose, and IRB-approved consent scripts will be presented to them. Observations will be conducted exclusively in those facilities where verbal consent has been obtained.

## Acknowledgments

We are thankful to the College of Medicine and Health Sciences, Hawassa University, for permitting us to conduct this study. We also acknowledge the contributions of Grammarly and Microsoft Copilot in enhancing the quality of the language of this manuscript. These AI tools provided helpful suggestions for improving the clarity, conciseness, and overall readability of the manuscript.

## Author contributions

**Conceptualization:** Melaku Haile Likka, Hiwot Abera Areru, Betelihem Eshetu Birhanu, Desalegn Tsegaw Hibistu, Bernt Lindtjørn.

**Formal analysis:** Melaku Haile Likka, Betelihem Eshetu Birhanu, Desalegn Tsegaw Hibistu, Bernt Lindtjørn.

**Funding acquisition:** Bernt Lindtjørn.

**Investigation:** Melaku Haile Likka, Hiwot Abera Areru, Betelihem Eshetu Birhanu, Desalegn Tsegaw Hibistu, Bernt Lindtjørn.

**Methodology:** Melaku Haile Likka, Hiwot Abera Areru, Betelihem Eshetu Birhanu, Desalegn Tsegaw Hibistu, Bernt Lindtjørn.

**Project administration:** Melaku Haile Likka, Hiwot Abera Areru, Bernt Lindtjørn.

**Supervision:** Desalegn Tsegaw Hibistu, Bernt Lindtjørn.

**Writing – original draft:** Melaku Haile Likka, Hiwot Abera Areru, Betelihem Eshetu Birhanu, Desalegn Tsegaw Hibistu, Bernt Lindtjørn.

**Writing – review & editing:** Melaku Haile Likka, Hiwot Abera Areru, Desalegn Tsegaw Hibistu, Bernt Lindtjørn.

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
