## [Decision Letter · Decision Letter 0]

30 Sep 2024

Dear Dr. Likka,

Thank you for submitting your manuscript to PLOS ONE. After careful consideration, we feel that it has merit but does not fully meet PLOS ONE’s publication criteria as it currently stands. Therefore, we invite you to submit a revised version of the manuscript that addresses the points raised during the review process.

**ACADEMIC EDITOR: Please insert comments here and delete this placeholder text when finished.** Be sure to:Be sure to:

Dear authors,

Thank you for writing this manuscript. It is of great interest to me.

Please revise your manuscript in terms of general English.

A revised version of your manuscript that takes into account the comments of the referee(s) will be reconsidered for publication.

Please note that submitting a revision of your manuscript does not guarantee eventual acceptance, and that your revision may be subject to re-review by the referee(s) before a decision is rendered.

Regards,

publication criteria and not, for example, on novelty or perceived impact. and not, for example, on novelty or perceived impact. and not, for example, on novelty or perceived impact. and not, for example, on novelty or perceived impact.

We look forward to receiving your revised manuscript.

Kind regards,

Habib Jalilian

Academic Editor

PLOS ONE

Additional Editor Comments:

Dear Authors,

Please answer all the comments raised by the referee.

The manuscript needs amendments in terms of general English.

Regards,

Reviewers' comments:

Reviewer's Responses to Questions

**Comments to the Author**

1. Does the manuscript provide a valid rationale for the proposed study, with clearly identified and justified research questions?

Reviewer #1: Partly

2. Is the protocol technically sound and planned in a manner that will lead to a meaningful outcome and allow testing the stated hypotheses?

Reviewer #1: Partly

3. Is the methodology feasible and described in sufficient detail to allow the work to be replicable?

Reviewer #1: Yes

4. Have the authors described where all data underlying the findings will be made available when the study is complete?

The PLOS Data policy requires authors to make all data underlying the findings described in their manuscript fully available without restriction, with rare exception, at the time of publication. The data should be provided as part of the manuscript or its supporting information, or deposited to a public repository. For example, in addition to summary statistics, the data points behind means, medians and variance measures should be available. If there are restrictions on publicly sharing data—e.g. participant privacy or use of data from a third party—those must be specified.requires authors to make all data underlying the findings described in their manuscript fully available without restriction, with rare exception, at the time of publication. The data should be provided as part of the manuscript or its supporting information, or deposited to a public repository. For example, in addition to summary statistics, the data points behind means, medians and variance measures should be available. If there are restrictions on publicly sharing data—e.g. participant privacy or use of data from a third party—those must be specified.

Reviewer #1: Yes

5. Is the manuscript presented in an intelligible fashion and written in standard English?

Reviewer #1: No

You may also provide optional suggestions and comments to authors that they might find helpful in planning their study.

Reviewer #1: Comments:

General: The protocol has picked an indispensable public health gap that requires prompt intervention and follow up.

Title: Very long and needs to be concise as “Improving the detection and management of common non-communicable diseases in adult population in Sidama National Regional State, Ethiopia: A study protocol”

Abstract: In the background section it is read as “PEN in Ethiopia is at an early stage and the readiness of rural primary healthcare units (PHCUs) to implement the strategy is unknown. We, therefore, propose to apply the strategy in the catchment areas of DobeToga Health Center, a rural PHCU in Sidama National Regional State (SNRS), Ethiopia, and improve the NCDs care among adults aged≥45 years. ”

I think the authors are not well aware of the Ethiopian Healthcare clinical guidelines (EPHCG) adopted by the ministry of health. The detection and management of hypertension and DM type 2 is well addressed. The problem is insufficient implementation of this guideline in primary healthcare units probably including in Sidama region. That is where you authors need to emphasize on.

I am also skeptical about the age group you are referring to (45 year and above) because there is age shift and younger persons are also acquiring the situation these days and the service should be rendered to all adults without segregation.

Aim: is too long and needs revision.

Methods in the abstract are not clearly narrated specifically in terms of tenses. Please distinguish what was already conducted and what will be conducted next.

Introduction: There are very long sentences difficult to conceptualize in each paragraph. The efforts already in place by the ministry of health to improve detection and management of NCDs have not been elucidated.

Objectives: There are redundant objectives like to determine the prevalence of undiagnosed diabetes etc…. (whatever the methods of detection are….)

There are no clear objective stated for the qualitative study designs.

Willingness to pay for glycolated hemoglobin …..why was it a special interest for the investigators when the most difficult part is to pay for drugs which is to be used lifelong than the investigation done periodically.

What about assessement and management of patients with multi-morbidity or co-morbidity?

Please focus on the key research questions as you are somewhat confused with many objectives…

Methods:

Please clearly indicate the study setting…. Is it about the catchment area of a health center or more than that to conduct your study?

It seems like you will be employing a mixed design…, Which mixed design, are you employing and why?

Would you please indicate your study population (both for qualitative and quantitative)? And how to select your study participants (sampling procedure) at each level has to be displayed schematically

The sample size allocated for different kebeles is not clear….what was the reason for having distinct sample sizes for different kebeles? Why was the design effect considered to be 2? Why not 1.5???

The language to be used for the interview sessions has not been clearly mentioned. The issue of using local languages translations (back and forward)

Operation definitions of key outcome variables are missing

What is the outcome variable for the cohort study and how will it measured and what will you be measuring as incidence and its predictors etc.????

Are you going to compute Kaplan Meier, cox regression etc.???

Improving patient ownership: Please try to capitalize on the concept of ‘person or patient centered care’ which is a contemporary concept to be dealt with as part of self-care and adherence to self-care.

The 31 items for self care need to be validated in Ethiopian context.. Please provide explanation on this. If not validated locally. It needs to be validated though you mentioned it as‘standardized’

It seems that you don’t have sample size for qualitative data…what will you be doing? And how will you do the analyses?

In your data analyses, what was the reason for employing R and stata softwares? Was that not possible to pick one preferable R as it is the most robust or powerful one?

Overall comment:

Do you think that your study will yield a different result as there are many studies with larger scope than yours on NCDs assessment and management in Ethiopia?

.

Reviewer #1: No

---

## [Author Response · Author response to Decision Letter 1]

4 Dec 2024

Academic Editor’s comments

Comment 1: “Please revise your manuscript in terms of general English.”

Responses: Thank you for your feedback. We have revised the language of the manuscript to improve clarity and readability. We utilized Microsoft Copilot and Grammarly to polish the language, reviewing and refining each paragraph to ensure the highest standard of English.

Comment 2: “A revised version of your manuscript that takes into account the comments of the referee(s) will be reconsidered for publication.”

Responses: We appreciate the valuable feedback provided by the referees. We have carefully revised the manuscript to address each comment and suggestion raised. Detailed responses to each referee's comments, along with a summary of the changes made to the manuscript, are provided below.

Comment 3: “Please ensure that your decision is justified on PLOS ONE’s publication criteria and not, for example, on novelty or perceived impact.”

Responses: We appreciate this comment. We have revised the resubmission to ensure that our protocol manuscript aligns strictly with PLOS ONE's publication criteria, which emphasize scientific rigor, ethical conduct, and methodological soundness. We have reviewed our study's objectives and methods to demonstrate their adherence to these criteria. We hope that these revisions address your concerns and look forward to your favorable consideration.

Comment 4:” Please include the following items when submitting your revised manuscript:

• “A rebuttal letter that responds to each point raised by the academic editor and reviewer(s). You should upload this letter as a separate file labeled 'Response to Reviewers'.

• “A marked-up copy of your manuscript that highlights changes made to the original version. You should upload this as a separate file labeled 'Revised Manuscript with Track Changes'.

• “An unmarked version of your revised paper without tracked changes. You should upload this as a separate file labeled 'Manuscript'.”

Responses: Thank you for your guidance. As requested, we have submitted the three items: this rebuttal letter, the track changed manuscript, and the manuscript with accepted changes (unmarked) according to your recommendations.

Comment 5: ““

Responses: We appreciate the suggestion. We have not made changes to our financial disclosure; thus, we have not updated the cover letter.

Comment 6 (journal requirements): When submitting your revision, we need you to address these additional requirements.

1. Please ensure that your manuscript meets PLOS ONE's style requirements, including those for file naming. The PLOS ONE style template can be found at https://journals.plos.org/plosone/s/file?id=wjVg/PLOSOne_formatting_sample_main_body.pdf and https://journals.plos.org/plosone/s/file?id=ba62/PLOSOne_formatting_sample_title_authors_affiliations.pdf

Responses: Thanks for letting us know. We have double-checked the revised manuscript to ensure it adheres to PLOS ONE's style guidelines, including file naming conventions. We've also referenced the style templates you provided.

Regarding ethical clearance, we have included an ethics statement in the Methods section (lines 517-544 in the revised manuscript). This statement outlines that ethical approval was obtained from the Institutional Review Board (IRB) of the College of Medicine and Health Sciences, Hawassa University (Ref. No: IRB/342/15). As the original protocol was modified, a revised ethical clearance was sought and approved by the Board (Ref. No: IRB/011/16). We also confirmed that participants were informed about the study's purpose and provided written consent, except for the quantitative data in Objective 5

Additional Editor’s comments

Comment 1: “Please answer all the comments raised by the referee.”

Responses: Thank you for your guidance. We have thoroughly addressed each comment raised by the referee. Our detailed responses are included in this rebuttal letter, along with references to the revised manuscript.

Comment 2: “The manuscript needs amendments in terms of general English.”

Responses: Thank you very much for this comment. We have revised the general English in the updated submission as outlined in the Response to the Editor’s Comment 1.

Reviewer’s comments

Comment 1: “Does the manuscript provide a valid rationale for the proposed study, with clearly identified and justified research questions?

“The research question outlined is expected to address a valid academic problem or topic and contribute to the base of knowledge in the field.

“Reviewer #1: Partly”

Responses: Thank you for your feedback regarding the rationale and research questions of our manuscript. We understand the importance of clearly identifying and justifying the research questions to address a valid academic problem and contribute to the field's base of knowledge.

In response to your comment, we have revised the manuscript to provide a more detailed rationale for the proposed study. We have also ensured that the research questions are more explicitly identified and justified, highlighting their relevance to addressing a significant academic issue. Specifically, we elaborated on the background and context of the study to better articulate the rationale. We also refined the research objectives in the revised submission.

We hope these revisions address the reviewer’s concerns. Thank you for helping us improve the manuscript.

Comment 2: “Is the protocol technically sound and planned in a manner that will lead to a meaningful outcome and allow testing the stated hypotheses?

“The manuscript should describe the methods in sufficient detail to prevent undisclosed flexibility in the experimental procedure or analysis pipeline, including sufficient outcome-neutral conditions (e.g. necessary controls, absence of floor or ceiling effects) to test the proposed hypotheses and a statistical power analysis where applicable. As there may be aspects of the methodology and analysis which can only be refined once the work is undertaken, authors should outline potential assumptions and explicitly describe what aspects of the proposed analyses, if any, are exploratory.

“Reviewer #1: Partly”

Responses: We appreciate your thoughtful feedback regarding the technical soundness of our protocol, as well as the comprehensive overview of our methods. We have revised the manuscript to address your concerns, ensuring that the methods are described with sufficient detail. Specific modifications to various elements of the methods are outlined in the "Review Comments to the Author" section below.

Comment 3: “Is the methodology feasible and described in sufficient detail to allow the work to be replicable?

“Descriptions of methods and materials in the protocol should be reported in sufficient detail for another researcher to reproduce all experiments and analyses. The protocol should describe the appropriate controls, sample size calculations, and replication needed to ensure that the data are robust and reproducible.

“Reviewer #1: Yes”

Responses: Thank you for your positive evaluation of the feasibility of our methodology and its detailed description.

Comments 4: Have the authors described where all data underlying the findings will be made available when the study is complete?

“The PLOS Data policy requires authors to make all data underlying the findings described in their manuscript fully available without restriction, with rare exception, at the time of publication. The data should be provided as part of the manuscript or its supporting information, or deposited to a public repository. For example, in addition to summary statistics, the data points behind means, medians and variance measures should be available. If there are restrictions on publicly sharing data—e.g. participant privacy or use of data from a third party—those must be specified.

“Reviewer #1: Yes”

Response: Thank you for your positive evaluation regarding this criterion

Comment 5: “Is the manuscript presented in an intelligible fashion and written in standard English?

“PLOS ONE does not copyedit accepted manuscripts, so the language in submitted articles must be clear, correct, and unambiguous. Any typographical or grammatical errors should be corrected at revision, so please note any specific errors here.

“Reviewer #1: No”

Responses: We appreciate your feedback regarding the language quality of our manuscript. We recognized the language issues and copyedited the entire manuscript, revising it paragraph by paragraph to enhance clarity and ensure it meets the highest language standards. We hope these revisions address your concerns and improve the overall clarity and readability of our manuscript.

Review Comments to the Author

R1 Comment 1: “General: The protocol has picked an indispensable public health gap that requires prompt intervention and follow up.”

Response: Thank you for your positive evaluation of our protocol.

R1 Comment 2: “Title: Very long and needs to be concise as “Improving the detection and management of common non-communicable diseases in adult population in Sidama National Regional State, Ethiopia: A study protocol”

Responses: We appreciate the suggestion. We accepted the recommended title with a minor modification. To signify the setting, we included the word “rural” in the title. The revised submission title is “Improving the detection and management of common non-communicable diseases in adults in rural Sidama National Regional State, Ethiopia: Study protocol.”

R1 Comment 3: “Abstract:

“In the background section it is read as “PEN in Ethiopia is at an early stage and the readiness of rural primary healthcare units (PHCUs) to implement the strategy is unknown. We, therefore, propose to apply the strategy in the catchment areas of DobeToga Health Center, a rural PHCU in Sidama National Regional State (SNRS), Ethiopia, and improve the NCDs care among adults aged≥45 years. “

“I think the authors are not well aware of the Ethiopian Healthcare clinical guidelines (EPHCG) adopted by the ministry of health. The detection and management of hypertension and DM type 2 is well addressed. The problem is insufficient implementation of this guideline in primary healthcare units probably including in Sidama region. That is where you authors need to emphasize on.

“I am also skeptical about the age group you are referring to (45 year and above) because there is age shift and younger persons are also acquiring the situation these days and the service should be rendered to all adults without segregation.

Responses: Thank you very much for this crucial comment. Following your suggestions, we reviewed the Ethiopian Primary Healthcare Clinical Guidelines (EPHCG) adopted by the Ministry of Health. We concurred with the reviewer's suggestions and have revised the manuscript accordingly. Specifically, we focused on the study area as reflected in lines 15-20 of the revised manuscript, changing from: “…However, the implementation of the PEN in Ethiopia is at an early stage, … ” to: “…However, the implementation of the PEN in rural health facilities in Sidama National Regional State is at an early stage….”

Regarding the age group our study is targeting, we recognize the increasing burden of DM type 2 and hypertension among younger individuals. However, for this study, the age group of 45 years and above was chosen due to financial and logistical constraints. We believe that focusing on this age group allows us to address a critical segment of the population at higher risk for the conditions. We will attempt to consider a broader age range to include younger individuals affected by these conditions in subsequent initiatives if resources allow. We appreciate your understanding of the constraints of this study and value your input for guiding potential future research directions.

R1 Comment 4: “Aim (in the Abstract): is too long and needs revision.

Responses: Thank you again for helping us improve our manuscript. We have revised the Aim (both in the Abstract (lines 22-26) and the body of the manuscript (lines 126-133)) to ensure it is more succinct while still capturing the essence of our study's objectives.

R1 Comment 5: “Methods in the abstract are not narrated specifically in terms of tenses. Please distinguish what was already conducted and what will be conducted next.”

Responses: We appreciate the comment, which helps improve the readability of the manuscript. In response to your feedback, we have ensured that the Methods (both in the Abstract and the body of the manuscript) are narrated clearly in terms of tenses to distinguish between completed and ongoing/future activities. Specifically, we have made the following updates:

- Phase 1 activities: We completed the phase 1 activities and collected data from the community-based screening. Currently, we are in the process of cleaning the data obtained from these community-based surveys.

- Phase 2 activities: We are currently implementing the second phase of the study. In this phase, we are applying the care model, collecting follow-up data, and measuring biochemical and physiological parameters. This phase is expected to be concluded by February 2025.

In the revised submission, we have improved the Methods section in both the Abstract and the body of the manuscript. We have reported the completed activities in the past tense and ongoing/future activities in the present and future tenses.

We hope these revisions enhance the clarity and readability of our manuscript.

R1 Comment 6: “Introduction: There are very long sentences difficult to conceptualize in each paragraph. The efforts already in place by the Ministry of health to improve detection and management of NCDs have not been elucidated.”

Responses: Thank you for your valuable feedback and for highlighting the need for further clarification on the efforts already in place by the Ministry of Health to improve detection and management of NCDs.

We have carefully reviewed the manuscript and identified the long sentences that may pose challenges for readers. We have revised these sentences to improve clarity and readability. Specifically, we have broken down complex sentences into shorter ones. These changes have been made throughout the manuscript to enhance the overall flow and comprehension.

Furthermore, to elucidate the efforts undertaken by the Ministry of Health, we have incorporated the following information into the revised submission: “Ethiopia has been implementing various measures to address the growing burden of NCDs, such as hypertension and diabetes. These measures include developing a national guideline for early detection and treatment and enhancing the capacity of primary healthcare units (PHCUs) in low-resource settings [2,21–23]. The PHCUs are crucial for addressing NCDs and providing integrated, equitable care. Enhancing the diagnostic capacity for NCDs is one of the strategic initiatives in Ethiopia's current national strategy for preventing and controlling cardiovascular and chronic diseases [24].” (lines 94-100)

We hope these revisions improve the clarity of the manuscript and adequately highlight the efforts made by the Ministry of Health.

R1 Comment 7: “Objectives: There are redundant objectives like to determine the prevalence of undiagnosed diabetes etc…. (whatever the methods of detection are….)”

Responses: Thank you for your feedback. We have taken your comments into account and revised the objectives. We eliminated redundant objectives and kept the relevant ones. Consequently, we have reduced the number of objectives from 14 in the original submission to 8 in the revised manuscript. (lines 134-156)

R1 Comment 8: “There are no clear objectives stated for the qualitative study designs.”

Responses: Thank you for your feedback. We do not have a distinct objective that

---

## [Decision Letter · Decision Letter 1]

18 Jul 2025

Dear Dr. Likka,

Thank you for submitting your manuscript to PLOS ONE. After careful consideration, we feel that it has merit but does not fully meet PLOS ONE’s publication criteria as it currently stands. Therefore, we invite you to submit a revised version of the manuscript that addresses the points raised during the review process.

We look forward to receiving your revised manuscript.

Kind regards,

Vincent Okungu, MPH, PhD

Academic Editor

PLOS ONE

Journal Requirements:

Reviewers' comments:

Reviewer's Responses to Questions

**Comments to the Author**

1. Does the manuscript provide a valid rationale for the proposed study, with clearly identified and justified research questions?

Reviewer #1: Partly

2. Is the protocol technically sound and planned in a manner that will lead to a meaningful outcome and allow testing the stated hypotheses?

Reviewer #1: Partly

3. Is the methodology feasible and described in sufficient detail to allow the work to be replicable?

Reviewer #1: Yes

4. Have the authors described where all data underlying the findings will be made available when the study is complete?

The PLOS Data policy requires authors to make all data underlying the findings described in their manuscript fully available without restriction, with rare exception, at the time of publication. The data should be provided as part of the manuscript or its supporting information, or deposited to a public repository. For example, in addition to summary statistics, the data points behind means, medians and variance measures should be available. If there are restrictions on publicly sharing data—e.g. participant privacy or use of data from a third party—those must be specified.requires authors to make all data underlying the findings described in their manuscript fully available without restriction, with rare exception, at the time of publication. The data should be provided as part of the manuscript or its supporting information, or deposited to a public repository. For example, in addition to summary statistics, the data points behind means, medians and variance measures should be available. If there are restrictions on publicly sharing data—e.g. participant privacy or use of data from a third party—those must be specified.

Reviewer #1: Yes

5. Is the manuscript presented in an intelligible fashion and written in standard English?

Reviewer #1: Yes

You may also provide optional suggestions and comments to authors that they might find helpful in planning their study.

Reviewer #1: Some of the concerns I have to the authors are:

1. As PEN apprpoach is still in its early stage of implementation in your study setting, ou may not be able to evaluate its performance. You may simply assess readniess of PHCU in detecting and managing common NCDs.

2. What was the reason for picking hypertension and diabetes why not other NCDs?

3. What was your oustanding reason to use qualitative method and how was the sample size computed or projected for the qualitative method needs further clarification. How were the data analysis done for qualitative method?

4. Why are you refering to Dobe Toga while collecting your data from 41 health centers and 4 primary hospitals? This has to be corrected.

5. Is the data collection still ongoing in the study settings. If yes, You may have a different result. I think the data collection has to be completed before analyzing your findings and organizing a manuscript.

.

Reviewer #1: No

---

## [Author Response · Author response to Decision Letter 2]

28 Aug 2025

Responses to the editor and the reviewer

Manuscript ID: PONE-D-24-13998R1

Title: Improving the detection and management of common non-communicable diseases in adults in rural Sidama National Regional State, Ethiopia: Study protocol

Journal: PLOS ONE

Dear Dr. Okungu and Reviewers,

We would like to thank you and the reviewers for the time and effort invested in reviewing our manuscript and for the constructive comments provided. We have carefully revised the manuscript to address all points raised. Changes made in the manuscript are highlighted in the tracked version submitted alongside this letter.

Reviewer’s comments and responses

Comment 1: “As PEN approach is still in its early stage of implementation in your study setting, you may not be able to evaluate its performance. You may simply assess readiness of PHCU in detecting and managing common NCDs.”

Response: We thank the reviewer for this insightful comment. We would like to clarify that our study does not evaluate the existing WHO PEN program, which is indeed in its early stage in our setting. Instead, we developed and implemented a WHO PEN–based care model for hypertension and diabetes diagnosis and management in a rural setting. Our study evaluates the effectiveness of this locally implemented model, rather than the original WHO PEN framework. This distinction has been clarified in the revised manuscript.

Changes made:

- Abstract (lines 22–23): Revised to: “We primarily aim to evaluate the effectiveness of a WHO PEN–based care model, which we implemented in a rural setting…” (previously: “We aim to evaluate the WHO PEN–based care model's effectiveness in controlling blood pressure and glucose levels among older adults.”)

- Study Aims section (lines 129–130): Revised to: “The study primarily aims to evaluate the effectiveness of a WHO PEN–based care model, which we implemented in a rural setting…” (previously: “We aim to evaluate the WHO PEN–based care model's effectiveness in controlling blood pressure and glucose levels among older adults.”)

In addition, the intervention we implemented is described in detail in both the previous and current submissions in the Methods section (lines 219-294).

Comment 2: “What was the reason for picking hypertension and diabetes? Why not other NCDs?”

Response: We appreciate this crucial question. Hypertension and diabetes were selected due to their high prevalence, public health importance, and feasibility of measurement within our study resources. These conditions are also prioritized under national and global NCD strategies, enhancing the potential policy relevance of our findings.

Changes made: - Introduction section, line 111-115 now includes a rationale for focusing on hypertension and diabetes over other NCDs

Comment 3: “What was your outstanding reason to use qualitative method and how was the sample size computed or projected for the qualitative method? How were the data analysis done for qualitative method?”

Response: Thank you very much for this valuable comment. We have clarified that the qualitative component is included to triangulate the quantitative findings on readiness and to provide a richer understanding of PHCU readiness to implement the WHO-PEN. These were described in Methods section of the submission. The sample size for qualitative interviews was determined based on the principle of data saturation, a widely accepted approach in qualitative research (described in the Sample Size Determination section, lines 323–327). Qualitative data were analyzed using thematic analysis, which involved transcribing the interviews and manually coding the data to identify key themes and patterns (explained in the Statistical Analyses section, lines 486–491).

Comment 4: “Why are you referring to Dobe Toga while collecting your data from 41 health centers and 4 primary hospitals? This has to be corrected.”

Response: We appreciate the reviewer’s careful reading of our manuscript. We would like to clarify that the primary objective of our study is to implement and evaluate the effectiveness of a WHO PEN–based care model, which was specifically implemented in Dobe-Toga Health Center. This is the reason why Dobe-Toga is mentioned in the context of the intervention.

The reference to 41 health centers and 4 primary hospitals pertains to a different component of our protocol, namely the assessment of the readiness of primary healthcare units in Sidama to implement the WHO PEN. This assessment was conducted across multiple facilities, which is why both Dobe-Toga Health Center and the broader set of PHCUs are included in the manuscript.

For clarity, the study is structured into multiple objectives: The intervention study on WHO PEN effectiveness, undiagnosed hypertension, and T2DM was conducted in Dobe-Toga Health Center. The readiness assessment and related health system studies were conducted in the 41 health centers and 4 primary hospitals in Sidama.

Comment 5: “Is the data collection still ongoing in the study settings? If yes, you may have a different result. I think the data collection has to be completed before analyzing your findings and organizing a manuscript.”

Response: Thank you very much for your comment. We confirm that data collection has been completed and currently, we are analyzing the data in accordance with the study objectives. As this is a study protocol, no study findings have yet been analyzed. We have clarified that the manuscript describes planned procedures and analysis, with results to be reported after completion of data collection.

We hope that these revisions satisfactorily address all reviewer concerns. We are grateful for the constructive feedback, which has improved the clarity and rigor of our study protocol.

Thank you for considering our revised manuscript for publication in PLOS ONE.

Sincerely,

Dr. Melaku Haile Likka

---

## [Decision Letter · Decision Letter 2]

18 Dec 2025

PLOS One

Dear Dr. Likka,

Thank you for submitting your manuscript to PLOS ONE. After careful consideration, we feel that it has merit but does not fully meet PLOS ONE’s publication criteria as it currently stands. Therefore, we invite you to submit a revised version of the manuscript that addresses the points raised during the review process.

<small>We recognise that the data collection for the study has now been completed and this has been a consequence of the time taken to review the manuscript. However, we require the protocol describing the study to be written in the future tense, as in the original version, so please can you re-write the text so that the manuscript describes the study that will be carried out (even if it has subsequently been completed).

Please note that if this is not undertaken appropriately then it may be necessary to reject your manuscript.</small>

We look forward to receiving your revised manuscript.

Kind regards,

Tope Michael Ipinnimo, MBBS, MPH, FWACP, FMCPH

Academic Editor

PLOS One

Journal Requirements:

Additional Editor Comments:

Thank you for submitting a revised version. We recognise that the data collection for the study has now been completed and this has been a consequence of the time taken to review the manuscript. However, we require the protocol describing the study to be written in the future tense, as in the original version, so please can you re-write the text so that the manuscript describes the study that will be carried out (even if it has subsequently been completed).

Please note that if this is not undertaken appropriately then it may be necessary to reject your manuscript.

Reviewers' comments:

Reviewer's Responses to Questions

**Comments to the Author**

1. Does the manuscript provide a valid rationale for the proposed study, with clearly identified and justified research questions?

Reviewer #2: Yes

2. Is the protocol technically sound and planned in a manner that will lead to a meaningful outcome and allow testing the stated hypotheses?

Reviewer #2: Partly

3. Is the methodology feasible and described in sufficient detail to allow the work to be replicable?

Reviewer #2: Yes

4. Have the authors described where all data underlying the findings will be made available when the study is complete?

The PLOS Data policy requires authors to make all data underlying the findings described in their manuscript fully available without restriction, with rare exception, at the time of publication. The data should be provided as part of the manuscript or its supporting information, or deposited to a public repository. For example, in addition to summary statistics, the data points behind means, medians and variance measures should be available. If there are restrictions on publicly sharing data—e.g. participant privacy or use of data from a third party—those must be specified.requires authors to make all data underlying the findings described in their manuscript fully available without restriction, with rare exception, at the time of publication. The data should be provided as part of the manuscript or its supporting information, or deposited to a public repository. For example, in addition to summary statistics, the data points behind means, medians and variance measures should be available. If there are restrictions on publicly sharing data—e.g. participant privacy or use of data from a third party—those must be specified.

Reviewer #2: Yes

5. Is the manuscript presented in an intelligible fashion and written in standard English?

Reviewer #2: Yes

You may also provide optional suggestions and comments to authors that they might find helpful in planning their study.

Reviewer #2: Thank you for the detailed manuscript. The previous reviews have been attended to except the last comment which is still not very clear.

1. The manuscript consistently employs past tense to describe completed procedures, such as “surveys were conducted” and “data were collected.”

I think Protocols are expected to outline planned methods and future procedures. The use of past tense implies that the protocol was prepared after data collection which could be concerning.

2. The manuscript lacks an explanation for the completion of quantitative data collection prior to protocol submission, whereas qualitative data collection has not yet commenced.

I think clarification of the study timeline is necessary, along with an explanation of how protocol integrity was preserved when a portion of the study was conducted before protocol submission.

3. Please can you specify the analytic approach for the qualitative data (such as thematic or content analysis), coding procedures and software that will be utilized.

.

Reviewer #2: **Yes:** Ayokunle Oluwadoyinsola AdedipeAyokunle Oluwadoyinsola AdedipeAyokunle Oluwadoyinsola AdedipeAyokunle Oluwadoyinsola Adedipe

---

## [Author Response · Author response to Decision Letter 3]

19 Jan 2026

We thank the editor and reviewers for their constructive comments. All comments have been carefully addressed in a point-by-point response provided in the uploaded “Response to Reviewers” document, and corresponding revisions have been made in the manuscript using track changes. We believe these revisions have significantly improved the clarity and quality of the manuscript.

---

## [Decision Letter · Decision Letter 3]

9 Mar 2026

Improving the detection and management of common non-communicable diseases in adults in rural Sidama National Regional State, Ethiopia: Study protocol

PONE-D-24-13998R3

Dear Melaku Haile Likka,

We’re pleased to inform you that your manuscript has been judged scientifically suitable for publication and will be formally accepted for publication once it meets all outstanding technical requirements.

Kind regards,

Tope Michael Ipinnimo, MBBS, MPH, FWACP, FMCPH

Academic Editor

PLOS One

Additional Editor Comments (optional):

Reviewers' comments:

Reviewer's Responses to Questions

**Comments to the Author**

1. Does the manuscript provide a valid rationale for the proposed study, with clearly identified and justified research questions?

Reviewer #2: Yes

2. Is the protocol technically sound and planned in a manner that will lead to a meaningful outcome and allow testing the stated hypotheses?

Reviewer #2: Yes

3. Is the methodology feasible and described in sufficient detail to allow the work to be replicable?

Reviewer #2: Yes

4. Have the authors described where all data underlying the findings will be made available when the study is complete?

The PLOS Data policy requires authors to make all data underlying the findings described in their manuscript fully available without restriction, with rare exception, at the time of publication. The data should be provided as part of the manuscript or its supporting information, or deposited to a public repository. For example, in addition to summary statistics, the data points behind means, medians and variance measures should be available. If there are restrictions on publicly sharing data—e.g. participant privacy or use of data from a third party—those must be specified.requires authors to make all data underlying the findings described in their manuscript fully available without restriction, with rare exception, at the time of publication. The data should be provided as part of the manuscript or its supporting information, or deposited to a public repository. For example, in addition to summary statistics, the data points behind means, medians and variance measures should be available. If there are restrictions on publicly sharing data—e.g. participant privacy or use of data from a third party—those must be specified.

Reviewer #2: Yes

5. Is the manuscript presented in an intelligible fashion and written in standard English?

Reviewer #2: Yes

You may also provide optional suggestions and comments to authors that they might find helpful in planning their study.

Reviewer #2: I have reviewed the responses to the previous comments and the corresponding revisions made to the manuscript. I am satisfied that all concerns raised in the prior rounds of review have been adequately addressed in this third revision.

Specifically, You have revised the manuscript to consistently describe all study procedures in the future tense, in line with protocol reporting standards,The explanation provided for the early completion of the quantitative component of Objective 5 prior to protocol submission is clear and acceptable. You have convincingly demonstrated that this did not compromise protocol integrity, as no data management, analysis, or interpretation was undertaken before protocol submission, and no modifications were made to the study design, objectives, or analysis plans.

Furthermore, the qualitative data analysis section has been substantially expanded to include the thematic analysis approach, coding procedures, and the software to be used, addressing my earlier concern about insufficient methodological detail.

I am satisfied with the revisions and recommend this manuscript for acceptance

.

Reviewer #2: No

---

## [Editor Report · Acceptance letter]

PONE-D-24-13998R3

PLOS One

Dear Dr. Likka,

I'm pleased to inform you that your manuscript has been deemed suitable for publication in PLOS One. Congratulations! Your manuscript is now being handed over to our production team.

Kind regards,

on behalf of

Dr. Tope Michael Ipinnimo

Academic Editor

PLOS One